 elife.elifesciences.org

# Developmental lineage priming in *Dictyostelium* by heterogeneous Ras activation

**Alex Chattwood[1†], Koki Nagayama[1†], Parvin Bolourani[2], Lauren Harkin[1], Marzieh Kamjoo[1], Gerald Weeks[2], Christopher RL Thompson[1*]**

[1]Faculty of Life Sciences, University of Manchester, Manchester, United Kingdom; [2]Department of Microbiology and Immunology, University of British Columbia, Vancouver, Canada

**Abstract** In cell culture, genetically identical cells often exhibit heterogeneous behavior, with only 'lineage primed' cells responding to differentiation inducing signals. It has recently been proposed that such heterogeneity exists during normal embryonic development to allow position independent patterning based on 'salt and pepper' differentiation and sorting out. However, the molecular basis of lineage priming and how it leads to reproducible cell type proportioning are poorly understood. To address this, we employed a novel forward genetic approach in the model organism *Dictyostelium discoideum*. These studies reveal that the Ras-GTPase regulator *gefE* is required for normal lineage priming and salt and pepper differentiation. This is because Ras-GTPase activity sets the intrinsic response threshold to lineage specific differentiation signals. Importantly, we show that although *gefE* expression is uniform, transcription of its target, *rasD*, is both heterogeneous and dynamic, thus providing a novel mechanism for heterogeneity generation and position-independent differentiation.

**\*For correspondence:** christopher. thompson@manchester.ac.uk

†These authors contributed equally to this work

**Competing interests:** The authors declare that no competing interests exist.

**Reviewing editor**: Janet Rossant, University of Toronto, Canada

## Introduction

Multicellular development requires the stereotypical and robust restriction of pluripotent cells to specific lineages. In many cases, this is dependent on positional information, where the relative position of a cell within the embryo determines the nature or amount of instructive differentiation signals received. However, there are also a growing number of examples of position independent patterning (*Kay and Thompson, 2009*). In these, different cell types firstly arise scattered in a 'salt and pepper' fashion before sorting out. To understand this mechanism, it will be important to understand why some cells differentiate, whereas neighboring cells within the same environment do not. One possible clue comes from cell culture studies that have revealed that genetically identical populations of cells exhibit heterogeneous behavior (*Chambers et al., 2007*; *Chang et al., 2008*; *Wu et al., 2009*). When these cells receive identical doses of defined differentiation inducing signals, only a small fraction of 'lineage primed' cells actually respond. In this scenario, a higher inducer concentration increases the number of responding cells without affecting the magnitude of the response of individual cells. This suggests that cells exhibit different intrinsic response biases or discrete transcriptional activation thresholds to signals.

There is now evidence to support the idea that the mechanisms underlying heterogeneous responses observed in cell culture could in fact regulate differentiation and developmental patterning in multicellular organisms (*Kaern et al., 2005*). For example, in one of the earliest lineage choices made during mouse embryogenesis, cells of the inner cell mass (ICM) adopt either primitive endoderm (PrE) or epiblast (EPI) fates. This occurs in a position independent fashion with ICM cells exhibiting

**eLife digest** How genetically identical cells develop into distinct cell types is one of the fundamental questions in biology. Certain molecules are known to act as signals that tell progenitor cells what type of cell they should become. The position of a cell within an embryo can determine which of these signals it is exposed to and thus influence its fate. However, it is also possible for a group of cells to be exposed to the same signal, but for only a few to respond. This gives rise to 'salt and pepper' differentiation—in which the cells differentiate in an apparently random manner to produce a mixture of different cell types—but the molecular basis of this phenomenon is unclear.

An organism called *Dictyostelium discoideum*, commonly known as slime mould, is often used to study these processes. *Dictyostelium* has an unusual life cycle; existing as individual cells when its bacterial food source is plentiful, with the cells coming together when food is scarce to form a multicellular slug that can move around. Cells within the slug turn into spores or into stalk cells, which lift the spores above the ground so that they can disperse. Under the right conditions, a single cell hatches from each spore; upon finding a new food source, this cell begins dividing thus allowing the life cycle to begin again.

The formation of stalk and spore cells occurs in a 'salt and pepper' pattern. A chemical messenger called DIF triggers cells to become stalk cells irrespective of their position within the aggregated mass of cells. Now, Chattwood et al. have shown that this process depends on the activity of two proteins; GefE and its substrate RasD. Surprisingly, both proteins are expressed many hours before cells differentiate, when cells are still well fed and dividing. Although GefE is uniformly expressed in these cells, its substrate—a protein called RasD—is expressed in only a subset of cells, and it is these cells that will later respond to DIF and ultimately become stalk cells.

The variable expression of RasD explains how 'salt and pepper' patterning arises following uniform exposure of apparently identical cells to DIF. It is likely that similar mechanisms have been conserved in higher organisms, so these findings could lead to a better understanding of how progenitor cells develop into specific cell types in multicellular plants and animals.

seemingly stochastic expression of PrE and EPI markers (*Dietrich and Hiiragi, 2007*; *Plusa et al., 2008*). It has been proposed that heterogeneity in responsiveness to differentiation inducing signals, such as the PrE inducer FGF, underlies this salt and pepper differentiation (*Yamanaka et al., 2010*). Crucially, in this model, it is not necessary for cells to receive different levels of FGF, only that they exhibit heterogeneity in their response thresholds to the signal. Finally, following this period of 'symmetry breaking', coherent tissues can emerge due to a process of sorting out. Sorting is likely caused by differential gene expression resulting in differential cell motility, which can be driven by chemotaxis or differential cell adhesion (with the elimination of misplaced cells also possible).

Pattern formation based on stochastic salt and pepper differentiation and sorting out is likely to be a fundamental and deeply conserved developmental patterning mechanism (*Kay and Thompson, 2009*). However, our knowledge of the underlying molecular mechanism, as to how heterogeneity affects responsiveness to differentiation signals, is still in its infancy. One route to understanding this phenomenon comes from the finding that initial cell fate choice and pattern formation in *Dictyostelium discoideum*, a genetically and biochemically tractable organism with a small number of easily recognizable cell types, is based on this mechanism (*Thompson et al., 2004b*). Upon starvation, *Dictyostelium* cells enter a developmental cycle that begins with the aggregation of several thousand cells to make a multicellular mound. Within the mound, cells differentiate intermingled into prestalk or prespore cells. After sorting out, the different cell types are organized into discrete tissues in the migratory slug. Prestalk cells occupy the anterior 25% of the slug, with anterior like cells also found scattered within the prespore zone. Upon culmination, prestalk cells undergo extensive rearrangements to populate distinct parts of the fruiting body, including the stalk and basal disc, as well as ancillary spore head supporting structures known as the upper and lower cup. Prestalk cells can further be subdivided into several major subtypes (pstA, pstO and pstB). Each cell type is defined largely by the pattern of expression of the classical prestalk specific transcripts *ecmA* and *ecmB* (*Jermyn et al., 1989*; *Williams et al., 1989*; *Jermyn and Williams, 1991*; *Early et al., 1993*), and by their position in the migratory slug and final fruiting body (*Figure 1—figure supplement 1*).

There is now good evidence to suggest that heterogeneities during growth of *Dictyostelium*, such as nutritional status or cell cycle position, affect developmental fate (*Leach et al., 1973*; *Gomer and Firtel, 1987*; *Thompson and Kay, 2000a*; *Chattwood and Thompson, 2011*). Firstly, cells grown without glucose preferentially become stalk cells when mixed in chimeric development with glucose grown cells (*Leach et al., 1973*; *Noce and Takeuchi, 1985*; *Blaschke et al., 1986*). Importantly, when cells from different growth conditions are compared, a hierarchy of biases results, where cells can be 'stalky' in one mixture but 'sporey' in another (*Leach et al., 1973*; *Thompson and Kay, 2000a*). Therefore, growth history does not commit a cell to a given fate, but is instead context dependent and relative to the growth history of the entire population. Secondly, biases established during growth have been shown to affect the sensitivity of cells to differentiation signals experienced during development, such as the chlorinated alkyl phenone DIF-1 (hereafter termed DIF), which is primarily required for pstB cell differentiation, with a more minor role in pstO cell differentiation (*Thompson and Kay, 2000b*; *Thompson et al., 2004a*; *Zhukovskaya et al., 2006*; *Huang et al., 2006b*; *Keller and Thompson, 2008*; *Saito et al., 2008*). For example, 'stalky' biased cells are more sensitive to DIF in monolayer culture assays than 'sporey' biased cells (*Thompson and Kay, 2000a*; *Kubohara et al., 2007*). Finally, immediate-early gene responses to DIF exhibit all-or-none behavior, leading to population level heterogeneity (*Stevense et al., 2010*). This provides a basis for salt and pepper differentiation in vivo, where it is presumed all cells experience similar DIF concentrations.

From these studies it is clear that in order to understand patterning by salt and pepper differentiation followed by sorting, it will first be important to understand why some cells respond to differentiation inducing signals whereas others do not. To address this, we have used a novel genetic approach to identify genes involved in lineage priming. Here we describe the characterization of one such gene, *gefE*, and show that its expression during growth is specifically required to determine the number of cells that respond to DIF, both in cell culture and in vivo. Finally, we show that although expression of *gefE* itself is not heterogeneous during growth, expression of its target, *rasD*, is both heterogeneous and dynamic. These studies thus establish how salt and pepper differentiation in response to a uniform diffusible signal can be achieved through the stochastic expression of a Ras gene, the activity of which in turn is dependent on a ubiquitously expressed regulatory Gef. These studies therefore provide key insights into the basis of gene regulatory networks that can underpin patterning during a poorly understood but fundamental mode of pattern formation.

## Results

### A genetic screen for modulators of lineage bias

The nutritional conditions under which cells are grown have previously been shown to bias lineage choice during *Dictyostelium* development (*Leach et al., 1973*; *Thompson and Kay, 2000a*). Consistent with these observations, we find that cells grown in the absence of glucose (G−) produce significantly fewer spores than cells grown in the presence of glucose (G+) when developed together in chimera (*Figure 1A*). Importantly, our analysis of the patterning of cell types at the slug and fruiting body stages reveals that this is because G− cells are biased towards the DIF dependent pstO and pstB lineages (*Figure 1B*, *Figure 1—figure supplement 2*). This is consistent with previous studies showing G− cells are more sensitive to DIF than G+ grown cells (*Thompson and Kay, 2000a*).

To better understand this lineage bias, we took a genetic approach in which a REMI mutant library estimated to initially contain approximately 1000 individual blasticidin resistant clones was hatched from a frozen stock two times. Each replicate pool was then grown separately in G− medium and developed in chimera with G+ wild type cells (*Figure 1C*). After each round of development, chimeric spore heads were transferred into growth medium containing blasticidin, thus killing all wild type cells and only allowing mutant spores to grow. This provides a selective pressure to enrich for mutants that adopt the spore cell fate. The selection was repeated on each pool for seven rounds. The success of the selection was assessed qualitatively by examining the distribution of mutant and wild type cells in chimeric slugs. As expected, in initial rounds, G− grown mutant cells were enriched in the pstO and pstB populations. However, in later rounds the distribution of mutant cells changed with a clear enrichment within the prespore zone, suggesting that mutants defective in bias imposed by G− growth were becoming enriched (*Figure 1—figure supplement 3A*).

Inverse PCR revealed the pool of cells in later rounds of each pool selection to be largely composed of clones in which the REMI plasmid had integrated into a small 5' 42 bp exon within the *gefE* gene

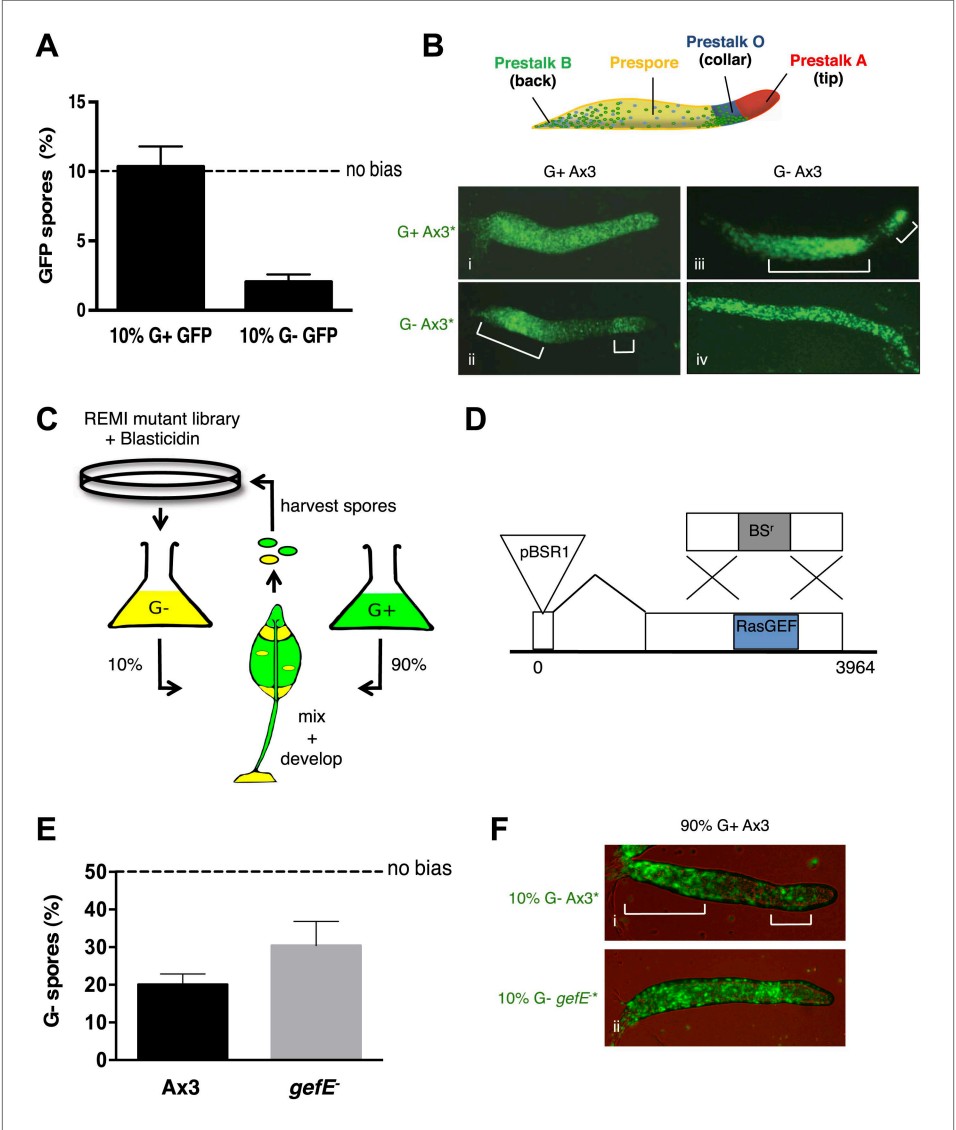

**Figure 1**. RasGEFE mutant cells are enriched in a genetic screen for modulators of nutritional bias.
(**A**) G− cells produce fewer spores than G+ cells in chimeric development. GFP-labelled Ax3 wild type cells were grown in either G+ or G− conditions and mixed 10:90 with wild type G+ cells. GFP spores were quantified by counting. Dotted line indicates the percentage GFP spores expected if there is no fate bias. Error bars represent SEM, p<0.0001. (**B**) G− growth biases cells towards pstO and pstB cell fates. Diagram shows organisation of different cell types along the anterior-posterior axis of the *Dictyostelium* slug. Patterning of GFP-labelled (*) G+ (i) and G− (ii) cells when mixed at 10:90 ratio with G+ cells. The reciprocal pattern was observed when GFP-labelled G+ (iii) and G− (iv) were mixed at 10:90 ratio with G− cells. (**C**) Schematic diagram of the genetic selection. REMI mutant cells were grown in G− and mixed 10:90 with wild type G+ GFP cells. Chimeric fruiting bodies were harvested and spores returned to growth medium after each developmental cycle. Wild type cells were removed with Blasticidin. (**D**) Generation of *gefE⁻* mutants. REMI plasmid, pBSR1, inserted into 42 bp exon 1. RasGEF catalytic domain (blue) deleted by homologous recombination. (**E**) *gefE⁻* mutant cells produce more spores than Ax3 wild type cells after G− growth. RFP-labelled wild type cells were grown in G+ medium and mixed at a 50:50 ratio with unlabelled wild type or *gefE⁻* mutant cells grown in G−. Number of unlabelled spores was quantified by counting. Error bars represent SEM, p<0.04. (**F**) Comparison of the patterning of GFP-labelled (*) wild type (i) or *gefE⁻* mutant (ii) cells grown in G− conditions when mixed at 10:90 ratio with unlabelled wild type G+ cells. AP axis in all slug images oriented from right-left with white bars showing regions of GFP enrichment.

*Figure 1. Continued on next page*

*Figure 1. Continued*

The following figure supplements are available for figure 1:

**Figure supplement 1**. Organisation of cell types in *Dictyostelim* slug and culminants.

**Figure supplement 2**. Cell type specific effects of nutritional history in *Dictyostelium* slugs.

**Figure supplement 3**. Enrichment of REMI mutants during screen correlates with change in cell fate preference during development.

---

(*Figure 1D*, *Figure 1—figure supplement 3B*). In one pool, *gefE* mutant cells were enriched by round 2, whereas in the second it was only strongly enriched by round 6. These differences could be due to the effects of random genetic drift caused by harvesting a finite number of spores after each round. Alternatively, it has recently been shown that when multiple clones are hatched and grown on bacteria, small differences in growth rate at the feeding edge can be amplified, leading to overrepresentation of some clones (*Buttery et al., 2012*).

We next performed a reconstruction experiment to verify that enrichment of *gefE⁻* mutant cells was due to being overrepresented in the spore population, rather than the mutation simply conferring advantages during other stages of the selection process. For this, because the catalytic domain of GefE is encoded by the larger 3' exon, we generated a mutant allele in which the catalytic domain was deleted (*Figure 1D*). For all subsequent studies, the catalytic domain disruption allele was used, but because both mutants exhibit identical phenotypes, both likely represent null alleles. Reconstruction experiments revealed no difference in the rate of growth of *gefE⁻* mutant cells in either G+ or G− medium, although *gefE⁻* mutant spores did, however, hatch at a faster rate than wild type spores, perhaps helping to explain the unexpectedly strong enrichment for this mutant within the pools (*Figure 1—figure supplements 3C,D*). Most importantly, when wild type or *gefE⁻* mutant G− grown cells were developed in chimera with G+ wild type cells, mutant G− cells were significantly overrepresented in the spore population of chimeric fruiting bodies (*Figure 1E*, p<0.04), thus providing a simple explanation for the enrichment of this mutant in the genetic selection. This is because mutant G− cells do not show the strong overrepresentation in the collar and back populations of chimeric slugs seen when wild type G− cells are mixed with wild type G+ grown cells (*Figure 1F*).

## *gefE⁻* mutant cells avoid DIF dependent prestalk cell fates

Growth in the absence of glucose specifically biases cells towards DIF dependent prestalk lineages. Because the *gefE⁻* mutant fails to respond to this G− growth bias, we next tested whether *gefE* gene disruption also resulted in specific defects in DIF dependent lineage bias. To test this, wild type and mutant cells were grown under identical growth conditions in the presence of glucose before mixing for chimeric development. Under these conditions, *gefE⁻* cells still formed more spores than wild type cells (*Figure 2A*, p<0.01). In chimeric slugs, labelled mutant cells were enriched in the prespore region and excluded from the DIF dependent pstB and pstO cell fates, whilst labelled wild type cells were localized in the pstB and pstO regions of the slug (*Figure 2B*, *Figure 2—figure supplement 1*). Similar results were seen when cells were grown in the absence of glucose (*Figure 2—figure supplement 2*). Finally, when GefE expression was driven by a strong constitutive promoter in wild type cells, cells expressing high gefE levels were found largely in the pstO and pstB regions of slugs and in the lower cup of fruiting bodies when mixed with wild type cells (*Figure 2C*). Together, these data suggest that expression of *gefE* is required for biasing cells to adopt DIF dependent pstO and pstB cell fates.

## GefE regulates DIF dependent lineage bias

Genes that affect cell fate choice can either affect lineage commitment or lineage bias. A lineage bias gene is one that is not required for cell fate commitment per se, but required for the *relative* probability of commitment to one fate over another. If the *gefE* gene is involved in lineage priming rather than commitment, an effect on development will not be observed if all cells in the population have the same bias (e.g., normal clonal development). We therefore examined cell fate choice and patterning of the *gefE⁻* mutant during clonal development, with all cells grown under identical conditions. Clonal development was found to be morphologically identical to that of the wild type. Furthermore, the number of prespore cells in *gefE⁻* mutant slugs was found to be the same as that observed in wild type

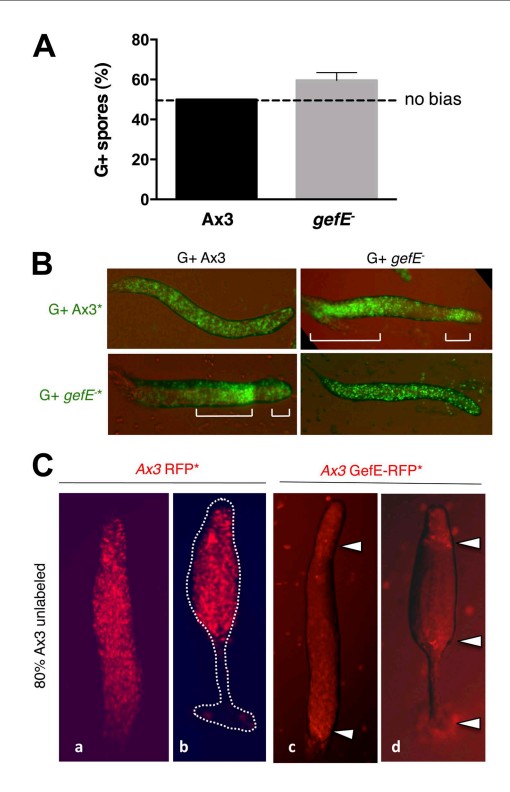

**Figure 2**. *gefE⁻* mutant cells avoid pstO and pstB fates. (**A**) RFP-labelled AX3 wild type cells were mixed at a 50:50 ratio with unlabelled wild type or *gefE⁻* mutant cells. Both strains were grown in presence of glucose. Number of unlabelled spores quantified by counting. Dotted line indicates the percentage RFP spores expected if there is no fate bias. Error bars represent SEM, p<0.01. (**B**) Chimeras of 10% GFP-labelled (*) wild type or *gefE⁻* mutant cells mixed with 90% unlabelled wild type or *gefE⁻* mutant cells. AP axis in all slug images oriented from right-left with white bars showing regions of GFP enrichment. (**C**) Cells transfected with RFP control vector (**a** and **b**), or GefE-RFP fusion vector (**c** and **d**), under control of the constitutive *actin* promoter, mixed with unlabelled wild type cells at 20:80 ratio and observed during slug (**a** and **c**) and culminant (**b** and **d**) stages of development. Closed arrows indicate relative enrichment in reporter gene expression.

The following figure supplements are available for figure 2:

**Figure supplement 1**. Cell type specific effects in chimeric *gefE⁻*/wt slugs.

**Figure supplement 2**. *gefE⁻* mutant cells avoid pstO and pstB fates when both partners have a G- growth history.

slugs, and the total number of spores produced in terminal fruiting bodies was the same for the *gefE⁻* mutant as for the wild type (*Figure 3A*). There was also no defect in prestalk cell specific gene expression in slugs or fruiting bodies (*Figure 3B*). The only difference between the clonal development of the *gefE⁻* cells and the wild type was a slight, although reproducible, delay in the initiation of aggregation (data not shown). These results are, therefore, consistent with the *gefE* gene being involved in lineage priming rather than lineage commitment. Furthermore, if the *gefE* gene is involved in lineage priming, rather than commitment, then the bias of the *gefE⁻* cells to avoid a DIF dependent fate, that is observed when they are mixed with wild type cells, should be reversed when they are mixed with cells of a stronger bias. Consistent with this idea, the lineage bias of the *gefE⁻* mutant cells was found to be dependent on the growth history of the partner genotype. For example, *gefE⁻* mutant cells were strongly biased *towards* DIF dependent fates when grown in the G− medium and then mixed with mutant cells grown in G+ medium (*Figure 3C*). These experiments provide strong evidence that *gefE⁻* cells are totipotent and that GefE regulates context dependent/relative DIF dependent lineage bias rather than cell fate choice.

## *gefE⁻* mutant cells are less sensitive to DIF

G+ grown cells are less likely to adopt DIF dependent lineages than G− cells and G+ cells have been shown previously to be less sensitive to DIF than G− grown cells (*Thompson and Kay, 2000a*). Therefore, we next tested whether *gefE⁻* mutant cells also exhibit differences in DIF responsiveness when compared to wild type cells. We found that *gefE⁻* mutant cells make fewer stalk cells than wild type across a 1000-fold range of DIF concentrations in the cAMP removal monolayer assay but exhibit similar sensitivity to changes in DIF concentration (*Figure 4A*). Importantly, growth in the absence of glucose results in a similar increase in DIF sensitivity in both wild type and mutant cells (*Figure 4A*), thus suggesting that GefE is only required to set basal levels of DIF sensitivity. Similar results were seen when the effect of *gefE* disruption was observed on the expression of primary DIF target genes. In the first experiment, cells were transformed with *ecmAO:lacZ* or *ecmB:lacZ* reporter genes. DIF induced gene expression of these reporters was reduced in the *gefE⁻* mutant compared to wild type at all DIF concentrations, but showed a

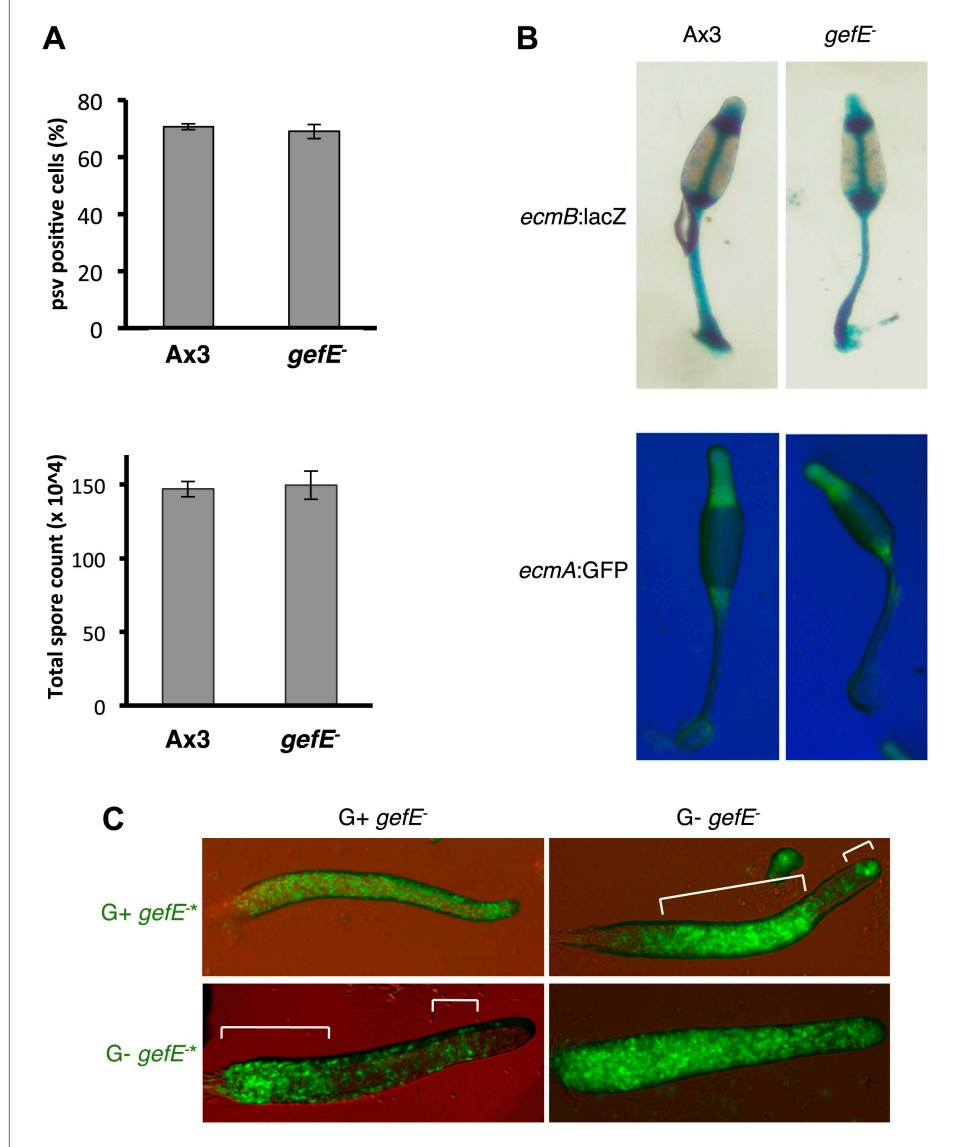

**Figure 3**. Cell type differentiation is unaffected during clonal development in the *gefE⁻* mutant. (**A**) Quantification of prespore:prestalk ratio and total number of spores produced by Ax3 wild type and *gefE⁻* mutant at slug and fruiting body stages respectively. (**B**) Expression of prestalk markers *ecmA* and *ecmB* in clonal Ax3 wild type and *gefE⁻* mutant culminants. (**C**) Chimeras of 10% GFP-labelled (\*) *gefE⁻* mutant cells grown in G+ or G− mixed with 90% unlabelled *gefE⁻* mutant cells grown in G+ or G− conditions. AP axis in all slug images oriented from right-left with white bars showing regions of GFP enrichment.

similar sensitivity to changes in DIF concentration (*Figure 4B*). A similar result was seen when endogenous *ecmA* and *ecmB* transcripts were measured in response to DIF induction (*Figure 4C*). Finally, we determined the effects of *gefE* disruption on an immediate-early DIF response. GATAc is a transcription factor that translocates to the nucleus minutes after DIF stimulation and is specifically required for DIF dependent pstB cell differentiation (*Keller and Thompson, 2008*). In *gefE⁻* mutant cells basal levels of nuclear GATAc were similar to wild type. Furthermore, when *gefE⁻* mutant cells were stimulated with DIF, the rate and duration of GATAc-GFP nuclear accumulation were largely unaffected. However, large differences were seen in the number of *gefE⁻* mutant cells that exhibit nuclear GATAc accumulation compared to wild type (*Figure 4D*). Together, these results suggest that *gefE⁻* cells are less sensitive to DIF than wild type cells, consistent with the idea that GefE biases cells towards DIF sensitive fates.

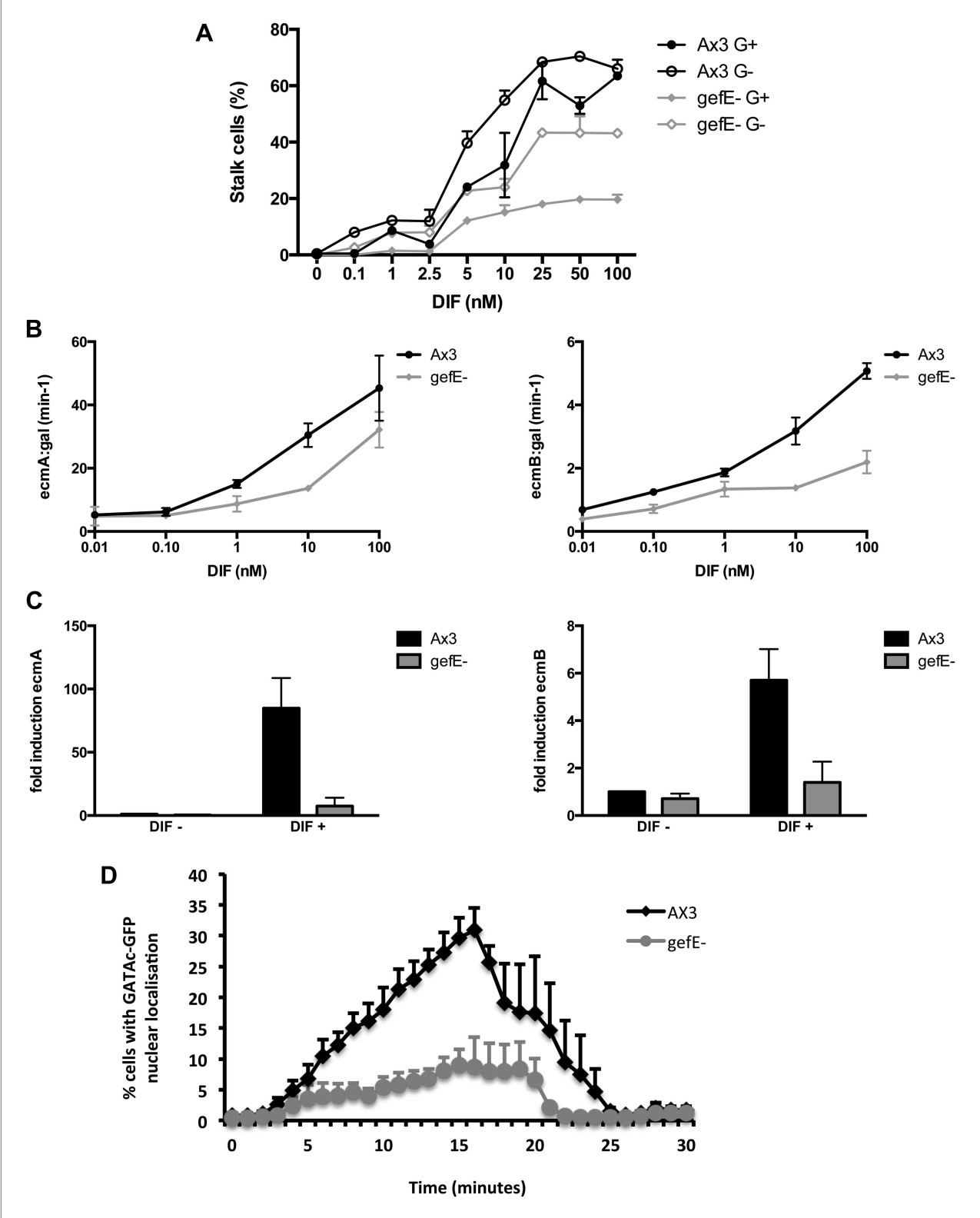

**Figure 4**. *gefE⁻* mutant cells are less sensitive to DIF. (**A**) Quantification of stalk cell formation 22 hr after DIF induction. (**B**) Expression level of lacZ reporter gene fused to prestalk specific promoter of *ecmA* (left) or *ecmB* (right) 22 hr after DIF induction. (**C**) qPCR analysis of endogenous *ecmA* (left) or *ecmB* (right) transcript levels after 3 hr induction with 100 nM DIF. (**D**) Nuclear translocation of DIF-induced transcription factor, GATAc-GFP in response to 100 nM DIF.

## *gefE* is required to set the DIF response threshold within individual cells

The above data demonstrate that *gefE⁻* mutant cells are less likely to respond to DIF. It has recently been shown that DIF induced transcription of the prestalk gene, *ecmA* only occurs once a cell intrinsic threshold is exceeded; increased DIF dose only results in a small change in target gene expression level in each cell, but a much larger increase in the number of cells that respond (*Stevense et al., 2010*). We, therefore, hypothesised that the reduced DIF responsiveness of the *gefE⁻* mutant was due to an increase in the threshold required to trigger a transcriptional response. To test this idea further, we developed a novel FACS based approach that permits *ecmA* expression levels to be measured in individual wild type or *gefE⁻* mutant cells in response to DIF. For this we replaced the endogenous copy of the *ecmA* gene with GFP (*Figure 5A*), and showed that GFP was expressed from the *ecmA* promoter in the same localization in slugs and fruiting bodies (*Figure 5B*) as has been previously described for *ecmA*-LacZ (*Jermyn and Williams, 1991*). As expected, in wild type controls, increasing DIF concentrations resulted in greater numbers of GFP expressing cells, with only a small change in

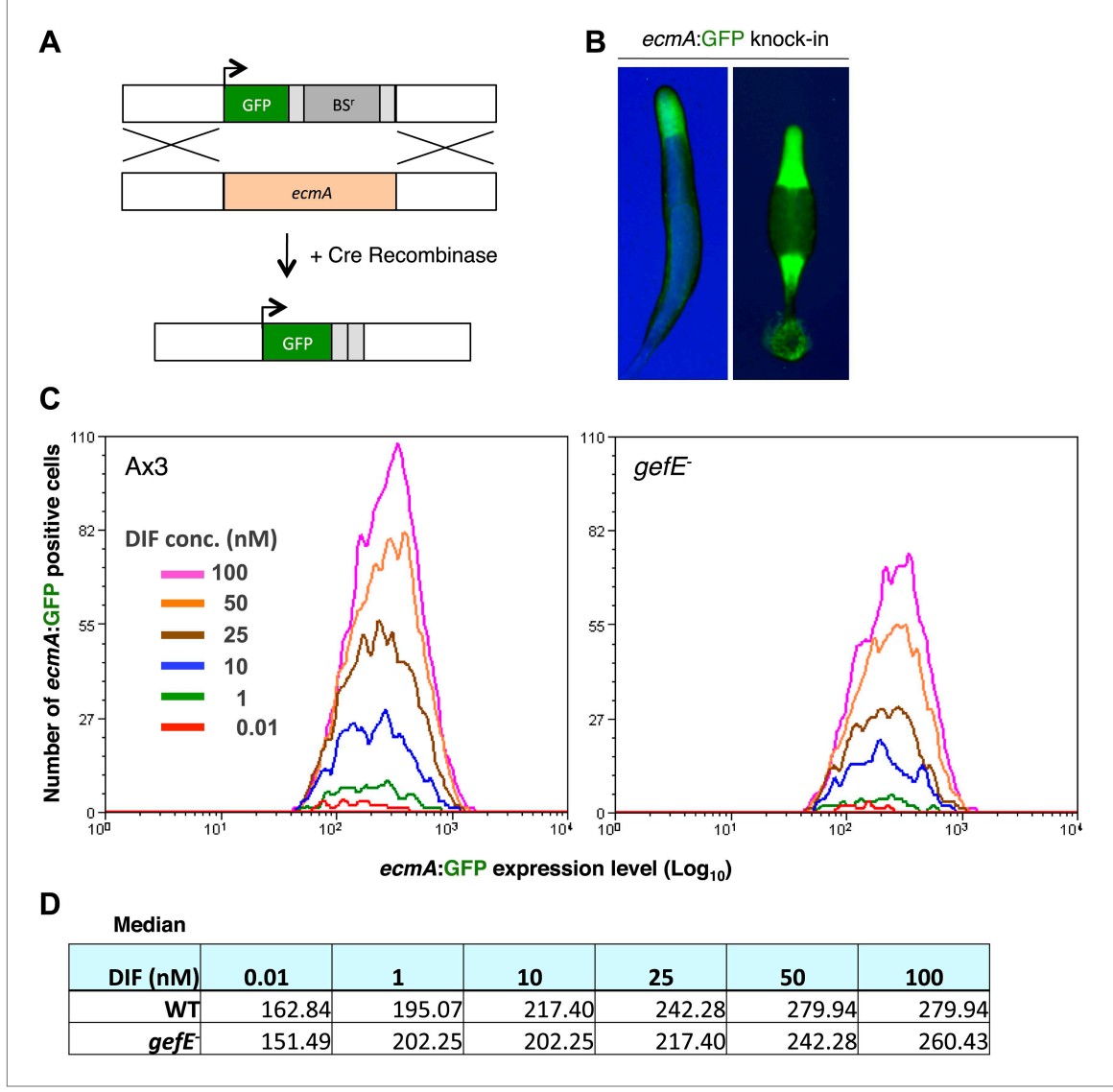

**Figure 5**. GefE regulates the DIF response threshold. (**A**) Replacement of the endogenous *ecmA* gene with GFP (**B**) prestalk specific expression of GFP knock-in strain at slug and culminant stages. (**C**) FACS analysis of Ax3 wild type and *gefE⁻* mutant GFP knock-in strains stimulated with 0.01–100 nM DIF for 9 hr. Y-axis shows the number responders and X-axis the GFP expression level per cell. (**D**) Median GFP expression level of wild type and *gefE⁻* mutant populations stimulated with 0.01–100 nM DIF.

GFP expression level per cell (*Figure 5C,D*). Most importantly, at all concentrations of DIF, the number of responding *gefE⁻* mutant cells was lower than wild type (*Figure 5C,D*). However, there was little difference in the level of *ecmA*:GFP expression between individual wild type or *gefE⁻* mutant responding cells (*Figure 5C,D*). Taken together, these data provide strong support for the idea that *gefE* regulates the cell-intrinsic threshold at which cells respond to DIF.

## GefE exerts its effects on cell type differentiation through RasD

We next sought to understand the molecular mechanism by which GefE affects lineage bias. As *gefE* is predicted to encode a RasGEF, whose primary function is to activate Ras proteins by catalyzing the exchange of GDP for GTP, it therefore seemed likely that GefE would exert its effects on DIF dependent lineage bias through Ras regulation. Although the *Dictyostelium* genome encodes 11 Ras proteins, only one, RasD, exhibits strong developmental expression (*Reymond et al., 1984*). Furthermore, we found that the level of both RasD expression and GTP bound activated RasD are significantly higher in cells grown in the absence of glucose (*Figure 6A*). Finally, GefE and RasD mutant cells have previously been shown to exhibit identical slug phototaxis defects (*Wilkins et al., 2000*, *2005*), raising the possibility that RasD could be a target for GefE. To test this idea directly, we firstly compared the relative levels of RasD-GTP in wild type and mutant cells by Western blot, using a pull down assay and a specific antibody (*Wilkins et al., 2000*). In 12 hr developed cells, the level of activated RasD was significantly lower in *gefE⁻* mutant cells, although there was no difference in the total level of the RasD protein (*Figure 6B*). In addition, there was no corresponding decrease in the levels of activated RasG, RasB, RasC or Rap1 (data not shown). These results suggest that GefE is the predominant activator of RasD and is specific for RasD. We therefore next tested whether *rasD⁻* and *gefE⁻* mutants exhibit similar cell fate choice phenotypes. When labelled wild type cells were mixed with *rasD⁻* mutant cells, the labelled cells accumulated in the pstO and pstB populations and were underrepresented in the prespore population (*Figure 6C*, *Figure 6—figure supplement 1*). Moreover, when RasD expression levels were elevated in wild type cells or cells were forced to express constitutively active *rasD*(G12T), these cells exhibited a strong localization towards the pstB region at the rear of slug and in the lower cup of fruiting bodies when mixed with wild type cells (*Figure 6D*). Importantly, unlike wild type cells, this effect was only observed in *gefE⁻* mutant cells which expressed constitutively active *rasD*(G12T). These findings thus suggest that RasD is normally converted to its GTP bound form upon activation by GefE and support the idea that GefE and RasD coordinate DIF dependent lineage bias.

## RasD expression is heterogeneous and dynamic in growing cells

Studies of the behavior of *gefE⁻* and *rasD⁻* mutant cells during chimeric development and in cell culture suggest that the levels of activated RasD are critical to set the DIF response threshold within populations of cells. This suggested that heterogeneous activity of RasD within a growing population of cells could perhaps underpin the salt and pepper differentiation of DIF dependent lineages during normal development. Therefore, we next tested whether *gefE* or *rasD* gene expression was heterogeneous during growth. For this, *gefE* or *rasD* promoter sequences were used to drive RFP expression. Although *gefE* was found to be uniformly expressed in all cells during growth, *rasD* expression was highly heterogeneous (*Figure 7A*). The *rasD* promoter RFP reporter construct used here shows strong developmental regulation and drives prestalk cell specific expression at the slug and culminant stages (*Figure 7—figure supplement 1*), consistent with previous reports using *rasD:lacZ* (*Esch and Firtel, 1991*; *Jermyn and Williams, 1995*) and observations of the endogenous transcripts (*Maeda et al., 2003*), validating its use as a measure of *rasD* promoter expression.

We next tested whether differences in *rasD* expression reflect differences in lineage bias. Firstly, we found that growth of cells in G− medium (conditions that increase the likelihood of DIF dependent lineage choice) resulted in a significant increase in the number of *rasD:RFP* expressing cells (73 ± 3% to 89 ± 5%) whereas no change was observed in *gefE:RFP* expressing cells. Secondly, growing *rasD:RFP* expressing cells were sorted by FACS based on their fluorescence and developed in chimera with wild type cells. Cells expressing lower levels of RasD were found to be less likely to differentiate as prestalk cells than prespore cells (*Figure 7B*). Surprisingly, however, cells expressing high *rasD* transcript levels can form both prespore and prestalk cells, as evidenced by their random distribution at both slug and culminant stages (*Figure 7B*). So, while there is an increased likelihood that high RasD cells will become prestalk cells compared to low RasD cells, they do not sort to pstO and pstB regions as might be expected.

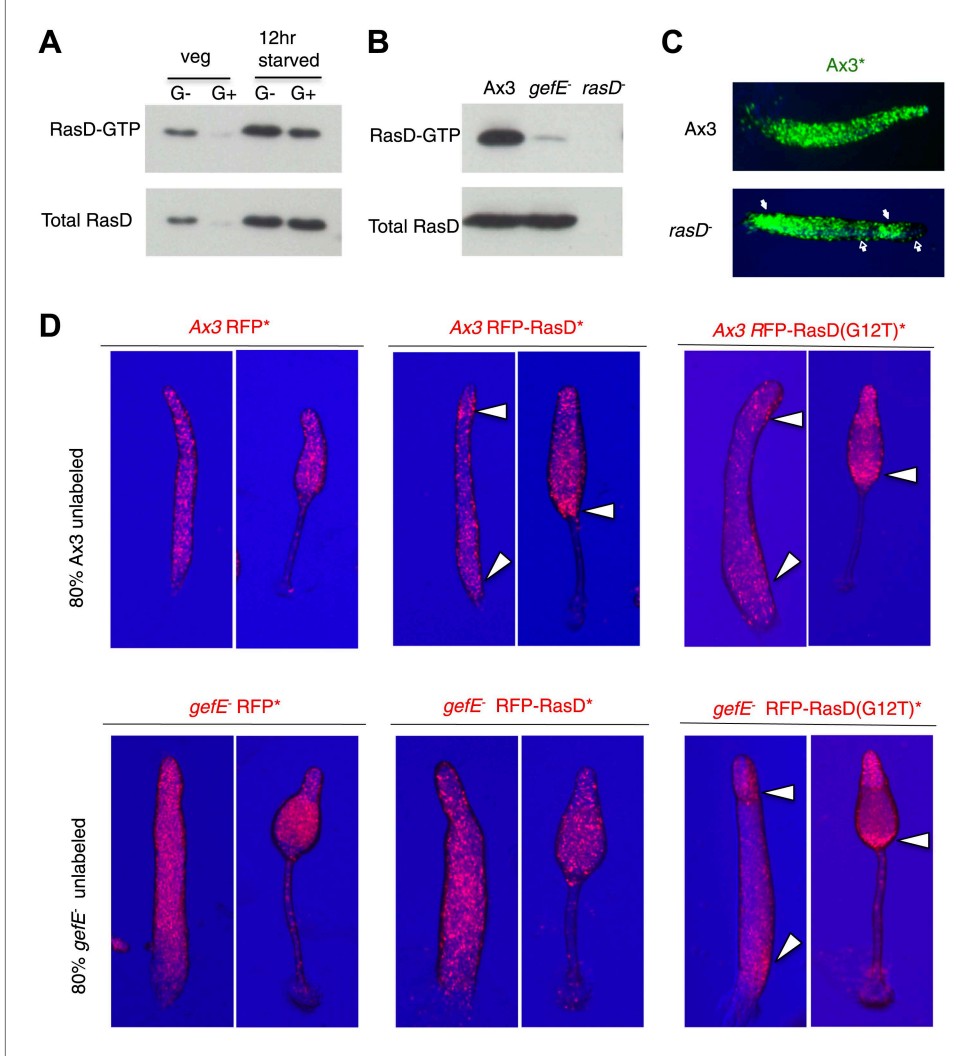

**Figure 6**. GefE activates RasD. (**A**) Comparison of the levels of activated RasD-GTP and total RasD by Western blot in vegetative or 12 hr starved cells grown in the presence or absence of glucose. (**B**) Comparison of activated RasD-GTP and total RasD levels by Western blot in wild type Ax3, *gefE⁻* or *rasD⁻* cells. (**C**) GFP-labelled (*) Ax3 wild type cells mixed at 10:90 ratio with unlabelled Ax3 or *rasD⁻* mutant cells. Closed arrows show enrichment of wild type cells in pstO and pstB populations. Open arrows show reciprocal enrichment of *rasD⁻* cells. (**D**) RasD overexpression results in GefE dependent bias towards the pstO and pstB cell fates. 20% cells constitutively expressing RFP, RFP-RasD or RFP-RasD(G12T) were mixed with 80% unlabelled parental cells. When wild type Ax3 cells overexpress RFP-RasD or RFP-RasD(G12T) they become enriched in the pstO and pstB populations (arrows). Only *gefE⁻* cells that express constitutively activated RasD(G12T) are enriched in pstO and pstB populations (arrows).

The following figure supplements are available for figure 6:

**Figure supplement 1**. *rasD⁻* mutant cells avoid pstO and pstB fates in chimera with wild type cells.

One likely explanation for this discrepancy is that the stability of the RFP protein may be different to endogenous RasD. For example, if RasD protein has a shorter half-life than RFP, then cells expressing RFP may no longer express RasD. These cells would thus not exhibit the expected prestalk bias and could differentiate as prespore cells.

Population heterogeneity in gene expression, such as that observed for *rasD* expression, can be achieved if gene expression fluctuates stochastically or in response to other intrinsic or extrinsic factors (**Raj and van Oudenaarden, 2008**; **Eldar and Elowitz, 2010**). This often leads to the establishment of a dynamic equilibrium that allows populations to return to normal after perturbation or respond

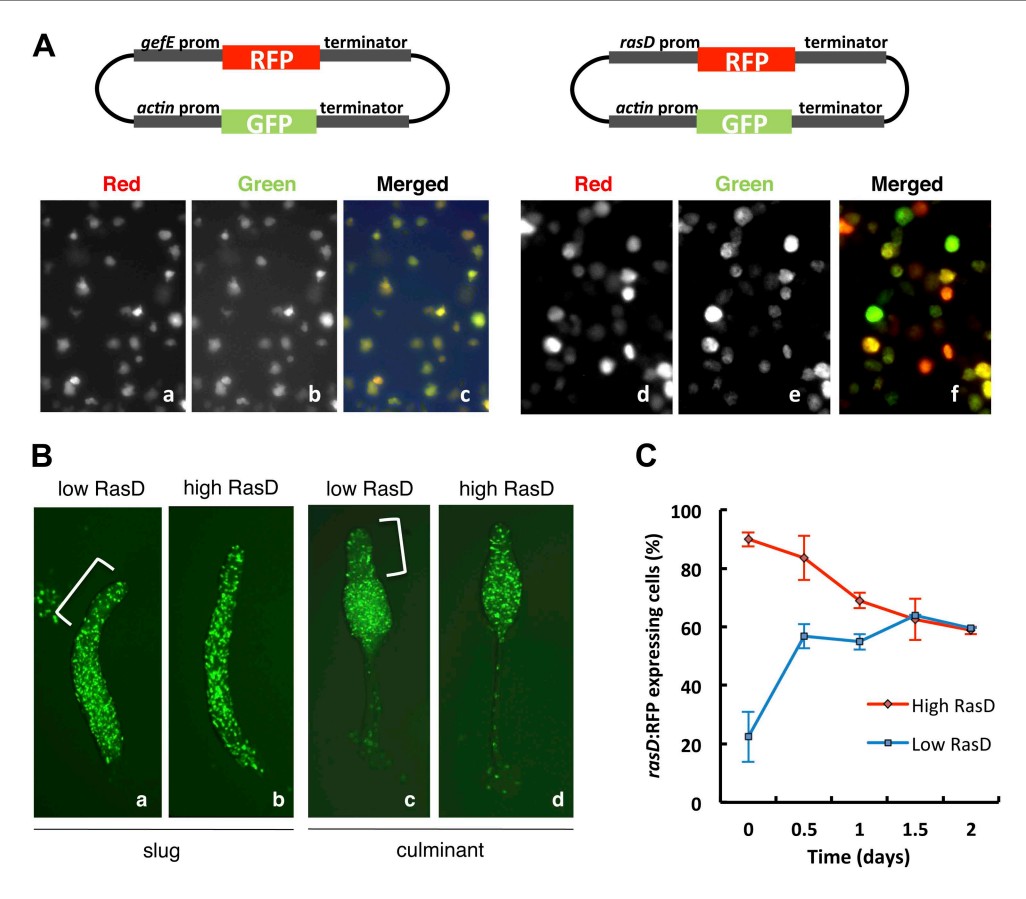

**Figure 7**. RasD expression is heterogeneous in growth phase populations. (**A**) Dual promoter vectors used to drive constitutive GFP expression and *gefE* promoter (left) or *rasD* promoter (right) driven RFP expression. Cells growing in tissue culture plates were photographed with a fluorescence microscope on red channel (**a** and **d**), green channel (**b** and **e**) and both (**c** and **f**). (**B**) Ax3 cells transformed with *rasD* promoter vector were fractionated into RFP high and RFP low populations by FACS. These populations were mixed in a 5:95 ratio with unlabelled Ax3 cells. Cell fate choice was traced by constitutive expression of GFP at slug (**a** and **b**) and culminant (**c** and **d**) stages. White bars show regions of fewer GFP cells. (**C**) FACS sorted low or high *rasD*:RFP cells were cultured back in HL-5 medium. The ratio of RFP:GFP cells was scored over time.

The following figure supplements are available for figure 7:

**Figure supplement 1**. RasD is expressed in prestalk cells during development.

appropriately to environmental cues (*Chang et al., 2008*; *Eldar and Elowitz, 2010*). In order to test whether RasD behaves in this fashion, *rasD* high or *rasD* low expressing cells were purified by FACS and then regrown in culture. Each population quickly returned to its pre-sorted equilibrium (*Figure 7C*). It is interesting to note, however, that the low RasD expressing population exhibits slightly faster dynamics than the high RasD expressing population, thus supporting the idea that chimeric behavior may be affected by differences in RasD transcription/translation and RFP stability. Together these findings suggest that population heterogeneity is generated by dynamic *rasD* expression in vegetative cells and that extrinsic factors such as nutritional history affect the position of this dynamic equilibrium.

## Discussion

The molecular mechanisms and gene regulatory networks controlling cell fate bias and lineage priming remain largely unknown. Our work, which identifies two bias genes, *gefE* and *rasD*, in *Dictyostelium* offers some insight. We find that transcription of *rasD* is heterogeneous in growing cells, whilst its activity

is controlled by a ubiquitously expressed RasGEF, *gefE*. Expression of GefE is required to activate RasD in a subset of cells, which in turn sets the threshold at which these cells respond to the pstB/pstO differentiation inducer DIF.

## Ras signaling and cell fate bias

Studies on the possible role played by *ras* genes in *Dictyostelium* development have not been conclusive (*Chubb and Insall, 2001*). Initial studies carried out in cell lines expressing constitutively activated Ras(G12T) suggested Ras signaling played a key role in prestalk cell differentiation (*Jaffer, 2000*; *Jaffer et al., 2001*). When these cells were developed in chimera with wild type cells, they were shown to occupy the collar and back of slugs, consistent with our present findings using RasD(G12T) (*Figure 6C*). However, disruption of the gene encoding RasD, the major developmentally expressed Ras protein, was reported not to affect cell fate specification during clonal development (*Wilkins et al., 2000*). This led to the suggestion that observations with constitutively activated Ras could be an artifact or that lack of RasD could be compensated by upregulation of other Ras genes (*Wilkins et al., 2000*). We found that there was indeed compensatory upregulation of the highly related RasG protein during development in both *rasD*⁻ and *gefE*⁻ mutant backgrounds (data not shown) but that this did not affect lineage bias in chimeras with wild type cells. This suggests that GefE-mediated activation of RasD is uniquely important to confer lineage bias in *Dictyostelium*. Such bias can be viewed as giving some cells either a head start or handicap in a 'race' to adopt fates. Only a limited number of 'winning' cells will differentiate before feedback mechanisms restrict other cells from adopting this fate, a process akin to lateral inhibition. Indeed, an analogous form of cell competition has been observed during *Drosophila* development when a clone of slow growing mutant cells is surrounded by wild type cells (*Levayer and Moreno, 2013*; *Vincent et al., 2013*). In this case, slow growing mutant cells are often eliminated by the faster growing wild type cells, which overgrow and compensate for their absence (*Simpson and Morata, 1981*). Like cell fate bias, cell competition is thus based on measurements of relative fitness between neighboring cells. Consequently, disruption of competition or bias genes only results in subtle defects during when all developing cells are the same genotype, such as a delay in the timing of cell fate determination.

## Heterogeneous Ras activation

Ras signaling has been the subject of intense study, with upstream and downstream components of Ras signaling pathways now well defined (*Karnoub and Weinberg, 2008*). However, we have less knowledge as to the possible importance of Ras regulation by transcriptional or post-translational control. Our studies using a reporter gene in which the *rasD* promoter drives RFP expression, reveals heterogeneous transcriptional activation of a *ras* gene within a population of growing *Dictyostelium* cells. Furthermore, this proportion increases when cells are transferred into medium lacking glucose. This indicates that there is a stochastic component to *rasD* gene expression that is influenced by external factors. In addition, when high or low *rasD* expressing cell populations are separated by FACS, they return to the pre-sorting equilibrium after a short period of growth. This suggests that cells freely transition between high and low *rasD* expressing subpopulations. Similar dynamic transcription has been observed in mammalian cell populations and is thought to represent slowly fluctuating transcriptomes that prime cell fate decisions both in the petri dish and during early embryogenesis (*Chang et al., 2008*; *Silva and Smith, 2008*; *Kalmar et al., 2009*; *Canham et al., 2010*; *Abranches et al., 2013*). Similarly, growing *Dictyostelium* cells expressing low levels of *rasD* transcripts are primed to become prespore cells during development, whilst RasD hyperactivation is sufficient to drive cells towards a prestalk cell fate. Surprisingly, however, we found that cells expressing high *rasD:RFP* transcript levels only exhibit a relatively weak prestalk preference. One interpretation of this finding is that if RasD protein has a shorter half life than RFP, then not all cells that express RFP would actually express RasD. Consequently, not all RFP expressing cells would exhibit a prestalk bias and may be free to differentiate as prespore cells, thus hindering experimental observation of their behavior. It is interesting to note, however, that similar observations of cell populations behaving more promiscuously in their lineage restriction than might be expected, have also been seen when fluorescent protein reporters are used in studies of the early mouse embryo and in embryonic stem cell cultures (*Canham et al., 2010*; *Grabarek et al., 2012*; *Morgani et al., 2013*). Most importantly, all these observations highlight the need for faithful real time reporters of gene expression, which will permit future studies of the regulation of RasD promoter activity, the molecular basis of transcriptional heterogeneity and lineage priming in general.

## DIF sensitivity and cell fate bias

Immediate-early DIF responses have been shown to be digital, with the number of responding cells being strongly affected by changes in DIF concentration, whereas the magnitude of the response in individual cells is largely unaffected (*Stevense et al., 2010*). This observation implies that (1) a response is only triggered once an internal threshold is exceeded and (2) the position of the threshold varies across a population. Work described in this report provides further support for this idea and gives mechanistic insight into how the DIF response threshold is set and, thus, how cell fate choice is regulated in vivo. Specifically, we find that GefE dependent heterogeneous RasD activation during the growth phase is required to set the normal DIF response threshold. However, several major questions emerge. For example, DIF is a developmental inducer that is present only after cells have been starved; yet the DIF response threshold is set (by RasD levels) before DIF exposure, in the growth phase. Indeed, heterogeneity in *rasD* expression is independent of DIF, as RasD expression is unaffected by DIF treatment (*Jermyn et al., 1987* and data not shown). One clue to how this heterogeneity is generated comes from our observation that RasD protein levels are higher in cells grown in medium lacking glucose. This suggests that heterogeneity in metabolic state between cells may play a role in *rasD* regulation. However, other growth phase heterogeneities, including differences in cell cycle position (*Gomer and Firtel, 1987*; *Araki et al., 1994*; *Thompson and Kay, 2000a*), intracellular calcium concentration (*Baskar et al., 2000*; *Azhar et al., 2001*) and intracellular pH (*Gross et al., 1983*; *Kubohara et al., 2007*) have all also previously been shown to affect cell fate choice. However, the molecular basis underlying these heterogeneities, or whether they affect RasD expression, is currently unknown. It thus remains to be determined whether they are a cause or consequence of the observed differences in RasD levels. Furthermore, it is currently unknown how RasD activation affects the sensitivity of cells to DIF at the molecular level. Because *gefE⁻* and *rasD⁻* cells are capable of forming DIF dependent cell types during development and *rasD* transcription is not induced by DIF (*Jermyn et al., 1987* and data not shown), these observations strongly suggest that RasD is not a central component of the DIF response pathway and that positive feedback through *rasD* transcriptional activation does not contribute to the binary switch. It thus appears that RasD only 'tunes' responses to DIF, rather than being integral to the DIF response per se. One clue to the molecular basis of this regulation, however, comes from the finding that the rapid (within 10 min) nuclear translocation of GATAc-GFP in response to DIF is affected in the *gefE⁻* mutant. Consequently, it seems likely that the tuning effects of RasD act during the immediate-early phase of the DIF response. However, to fully understand how this tuning of DIF sensitivity occurs will require a greater knowledge of the DIF signaling pathway itself, and crucially how each component is regulated (as all components represent potential RasD targets). For example, to date, only a small number of transcription factors and one protein kinase have been shown to be crucial for DIF signal transduction and responses (*Fukuzawa et al., 2001*; *Thompson et al., 2004a*; *Zhukovskaya et al., 2006*; *Huang et al., 2006a*; *Keller and Thompson, 2008*; *Araki et al., 2012*). However, the regulation of their activity is complex, requiring control by posttranslational modifications, as well as subcellular localization (*Fukuzawa et al., 2001*, *2003*; *Thompson et al., 2004a*; *Zhukovskaya et al., 2006*; *Huang et al., 2006a*; *Araki et al., 2008*, *2012*; *Keller and Thompson, 2008*). It is therefore possible that RasD alters the activity of specific protein kinases or transcription factors or hitherto unknown DIF signaling components, making them more likely to be DIF induced. In this way, because the DIF signaling pathway is intact in *rasD⁻* and *gefE⁻* mutant cells, they would still respond to DIF (albeit less efficiently than wild type) when given prolonged exposure to the signal. Finally, it is unclear to what extent inputs from different signaling pathways are also integrated to modulate the sensitivity of cells to DIF in response to nutritional bias, or other modulators of cell fate bias. Indeed, it is likely that the effects of nutrition are only partially explained by GefE activity. For example, when *gefE⁻* mutant cells are grown in the absence of glucose are compared to wild type cells grown in the presence of glucose, they do not produce as many spores in chimeric development. Furthermore, the *Dictyostelium* retinoblastoma ortholog, RblA, is a cell cycle regulated protein whose expression level during the growth phase is correlated with both DIF sensitivity and cell fate preference during development (*MacWilliams et al., 2006*). Interestingly, *rblA⁻* mutant cells display hypersensitivity to DIF and adopt collar and back cell fates when mixed with wild type cells in chimera, perhaps indicating a degree of antagonism between Ras and Retinoblastoma signaling. Similarly, knock out of the Glycogen Synthase Kinase 3 homolog, *gskA*, results in DIF hypersensitivity and the *gskA⁻* strain preferentially forms basal disc cells at the expense of spores when developed

clonally (*Harwood et al., 1995*; *Schilde et al., 2004*). Such similarities, as well as clear phenotypic differences (data not shown), suggest this level of complexity allows the DIF signal to be integrated into multiple fate choices during development.

### *Dictyostelium* as a model for lineage priming

Development in *Dictyostelium* is unusual, being based on aggregation of separate cells, rather than division of a fertilized egg. However, remarkable similarities are emerging between this evolutionarily ancient organism and the behavior of embryonic stem cell cultures, and indeed lineage specification in the early mouse embryo. (1) Salt and pepper differentiation of prestalk and prespore cells: During mouse embryogenesis, the PrE and EPI lineages arise from the ICM. Each lineage can be clearly defined at later stages through specific gene expression profiles, as well as their position within the blastocyst. However, at the earliest stages cells expressing PrE and EPI lineage markers are intermingled, with active sorting out observed at later stages (*Dietrich and Hiiragi, 2007*; *Plusa et al., 2008*; *Yamanaka et al., 2010*). (2) Dynamic expression of lineage priming/specific gene expression: observations in cell culture reveal expression of ICM cell fate markers such as Hex (*Canham et al., 2010*) and Nanog (*Chambers et al., 2007*; *Kalmar et al., 2009b*) is highly heterogeneous. When high or low level expressing cells are separated by FACS, the population structure returns to an equilibrium on time-scale of hours to days. Most importantly, differences in expression levels also correlate well with fate preferences in chimeric development (*Canham et al., 2010*). (3) Modulation of Ras activation affects lineage choice: although FGF activates multiple pathways, genetic studies indicate that the Grb2 > Sos (RasGEF) > Ras > Erk axis is the most important during differentiation of both ES cells and PrE (*Chazaud et al., 2006*; *Lanner and Rossant, 2010*). (4) Heterogeneous responses/thresholds to differentiation signals: FGF signaling is required for normal EPI/PrE differentiation (*Yamanaka et al., 2010*). However, it has recently been shown that salt and pepper expression pattern of EPI and PrE markers does not require FGF signaling (*Kang et al., 2013*). Rather, activation of this pathway is associated with exit from pluripotent state and commitment to the PrE fate. Finally when FGF levels are manipulated, the proportion of PrE:EPI cells correlates well with FGF levels, implying existence of cell intrinsic threshold to FGF (*Yamanaka et al., 2010*; *Grabarek et al., 2012*).

### Understanding the molecular basis of lineage priming

While it is clear that multicellular development requires extreme precision, there is a growing realization that key decisions are biased by heterogeneous behavior between equipotent cells. Population level variation in sensitivity to a global signal is one way to generate diversity. However, our understanding of the mechanistic basis of lineage priming is still in its infancy. One reason for this is that mutants in lineage priming genes may only show subtle defects during clonal development. Importantly, our study highlights that testing chimeric effects can uncover important regulatory roles for these genes. Due to the relative ease with which chimeric development can be assayed in *Dictyostelium*, it seems likely that studies in this model system will lead to a better understanding of how lineage biases are acquired and potentially how they might be manipulated to increase the efficiency of stem cell based therapeutics.

## Materials and methods

### Cell culture and development

*Dictyostelium* strains were cultured on lawns of *Klebsiella aerogenes* or in HL5 medium with (G+) or without (G−) 86 mM glucose. Cells were grown in shaken suspension for 2–4 days for G− phenotypes. All cultures were maintained at log phase ($1–4 \times 10^6$ cells/ml) during this period. Cells transformed by electroporation were selected with blasticidin (10 µg/ml) or G418 (20–40 µg/ml). For development, amoebae were washed with KK2 (16.1 mM $KH_2PO_4$, 3.7 mM $K_2HPO_4$) and deposited onto KK2 plates containing 1.5% purified agar (Oxoid) at a density of $3.5 \times 10^6$ amoebae/cm². Plates were kept for 14–16 hr at 22°C in a dark, moist box then removed and allowed to complete development in the light.

### Genetic selection for bias mutants

REMI mutagenesis (*Kuspa and Loomis, 1992*) was performed as described in *Parkinson et al., 2011*. 25 individual transformations producing approximately 1000 mutant clones were pooled after 4 days growth in 10 µg/ml blasticidin. This library of clones was used to perform two selections in parallel. Mutant G− cells were mixed with GFP labelled wild type G+ cells at a 10:90 ratio and developed

to the mature fruiting body stage. Spores were collected first into spore buffer (10 mM EDTA, 0.1% NP-40) and then into HL5 +86 mM glucose. After hatching, amoebae were incubated with 10 μg/ml blasticidin for 4 days to remove wild type cells and enrich for mutants. This selection was repeated for seven rounds. Mutant loci were identified by iPCR (*Keim et al., 2004*).

## DIF assays

Stalk cell differentiation was quantified using the cAMP removal described in *Thompson et al., 2004a*. Expression of *ecmA:lacZ* and *ecmB:lacZ* reporter genes was induced by addition of 0.01–100 nM DIF-1 and 50 μM cerulenin to stalk medium (10 mM MES, pH6.2, 1 mM CaCl₂, 2 mM NaCl, 10 mM KCl, 200 μg/ml streptomycin sulphate) containing 5 mM cAMP. After 22 hr incubation at 22°C, levels of lacZ in cell lysates measured as described in *Parkinson et al., 2011*. Endogenous levels of *ecmA* and *ecmB* after 9 hr induction with 0.01–100 nM DIF were measured by qPCR as described in *Huang et al., 2006*. Nuclear translocation of GATAc-GFP in response to 100 nM DIF was measured as described in *Keller and Thompson, 2008*.

Individual-cell responses to DIF quantified by knocking in a single copy of GFP at the endogenous *ecmA* locus. Cells were incubated in stalk medium + 5 mM cAMP for 9 hr. Then, another 200 μl dose of 200 mM cAMP was added with 0.01–100 nM DIF. After a further 9 hr incubation, cells were resuspended in 1 ml KK2 and analysed on a Beckman Coulter Cyan ADP FACS machine.

## RasD expression level and cell fate choice

Exponentially growing *rasD*:RFP promoter cells were subjected to FACS analysis. Cells expressing low levels of *actin*:GFP were removed and high and low RasD expressing cells were collected. Those cells were mixed with unlabelled wild type cells in a 5:95 ratio and developed in chimera. To measure dynamics of RasD expression, FACS sorted low or high *rasD*:RFP cells were returned to HL-5 medium. As the numbers of recovered cells was quite low, unlabelled wild type cells were added to ensure exponential growth. The number of RFP/GFP expressing cells was scored every 12 hr for 2 days.

## Qualitative and quantitative observations of cell fate

Developmental structures were observed under a Leica S6D dissection microscope or a Leica MZ16FA stereoscope. Whole-mount lacZ staining performed as described in *Dingermann et al., 1989*. For measurement of prespore:prestalk ratio, dissociated slug stage cells were disaggregated in KK2/20 mM EDTA with 21 G needle. Cells were then fixed and stained with prespore-specific anti-psv antibody as described in *Forman and Garrod, 1977*. For measurement of total spore number, $2 \times 10^6$ cells in 20 μl were spotted onto KK2 agar and incubated at 22°C in the dark for 2 days. Spores were harvested in KK2 with 0.1% NP40/20 mM EDTA and scored with a hemocytometer.

## Knockout constructs

The *gefE⁻* mutant allele (*Wilkins et al., 2005*) was obtained from the Dictybase Stock Center. In this study, two additional *gefE⁻* mutant alleles were generated. To disrupt the catalytic domain, a 2.1 kb fragment of the *gefE* gene was amplified by PCR (5′-GTAATGGCTCGAAAGTCCTTC and 5′-TTAAGAC TTAAAAGAATT) and cloned into the TOPO pcr2.1 vector. A floxed blasticidin resistance cassette derived from pLPBLP by Sma1 digestion was then inserted by blunt ligation at the endogenous Bst171 site in the catalytic domain of *gefE*. A second construct was made to delete a region of 855 bp from the catalytic domain. Two fragments of the *gefE* gene incorporating restriction sites were amplified by PCR (gefE_sal–AGGC*GTCGAC*CACCCTATAGTCCAGATAC, gefE_nco–AGG*CATCGG*AATCTCTGTTG GATC, gefE_pst–ATG*CTGCAG*CAGTTTCTGATCGACC, gefE_not–ATA*GCGGCCGC*CGTTGATTATGA GCAT) and cloned into pLPBLP, one on each side of the floxed blasticidin. The *rasD⁻* mutant allele described in this paper was recreated with the pATW1 construct used to generate the original *rasD⁻* knockout (*Wilkins et al., 2000*).

## Expression constructs

For constitutive reporter gene expression, pDM318 or pDM324 (*Veltman et al., 2009*) were used. For cell type specific expression studies pDd19 (pspA), pEcmAO-gal and pEcmB-gal vectors (*Jermyn and Williams, 1991*) were used. Reporter genes were cloned into these vectors as required. The complete 3.1 kb *gefE* coding sequence was amplified from wild type cDNA by designing primers (gefEF BamHI EEtag–CCC*GGATCC*AAAATGGAATATATGCCAATGGAAATGGATCATACTGAGTGTAAC, gefER SpeI-AATT*ACTAGT*AGACTTAAAAGAATTTAAAAG) and cloned into pDM324 with the appropriate

restriction sites. The complete *rasD* gene was amplified from wild type gDNA by designing primers (rasDF BglII EEtag–CCC*AGATCT*AAAATGGAATATATGCCAATGGAAACAGAATATAAATTAGTTATT GTAGG, rasDR SpeI–CCCC*ACTAGT*TAAAATTAAACATTGTTTTTTC) and cloned into pDM318 with the appropriate restriction sites. Using this vector as a template, primers were designed (G12TF–GTTATTGTAGGTGGTACTGGTGTTGGTAAAAGTGC, and G12TR–GCACTTTTACCAACACCAGTA CCACCTACAATAAC) to generate the *rasD*(G12T) vector by PCR using a QuikChange I Site-Directed Mutagenesis Kit (Agilent Technologies, UK).

For *gefE* and *rasD* promoter studies, 900 bp or 700 bp of upstream sequence plus the first 7 or 13 codons, respectively, were amplified by designing primers (gefEpF XhoI–AATTCTCGAGCAAAATT GATTGTAAAGCTGG, gefEpR BglII–AATTAGATCTGTTACACTCAGTATGATCCAT, rasDpF XhoI–ACTG*CTCGAG*TCTTATAATTTGGTTAAATCGATG, and rasDpR BglII–AAA*AGATCT*ACCACCACCACC TACAATAAC) and cloned into pDM324 with appropriate restriction sites. To ensure variation in RFP expression was specific to *gefE* or *rasD* promoters, *actin* promoter-driven GFP was cloned from pDM327 (*Veltman et al., 2009*) into *Ngo*MIV site of each vector.

### *ecmA* promoter: GFP knock-in

First, the full length of GFP gene was amplified from pTX-GFP (*Levi et al., 2000*) by PCR (GFPf_SalI–acgc*GTCGAC*GGAACCAGTAAAGGAGAAGAAC and GFPr_stop_HindIII–acgc*AAGCTT*TTATGCATC TCGAGTGGAACC) and cloned into *Sal*I/*Hind*III sites of pLPBLP (*Faix et al., 2004*). Then, two 2000 bp fragments of the upstream sequence plus the first codon and the downstream sequence including the stop codon of the *ecmA* gene were amplified from wild type genomic DNA by designing primers (ecmA_up_F_KpnI acgc*GGTACC*ATAGTCGATAATTTCATTACATCATC, ecmA_up_R_SalI acgc*GTCGAC* CATTTTCAACGTTATAATTTTTAAAC, ecmA_down_F_PstI acgc*CTGCAG*TAAATAACTCTTTTTATT TAATTATATTTT, and ecmA_down_R_BamHI acgc*GGATCC*TACGTATTGAAATTCATCATCC). These PCR products were cloned into the GFP integrated pLPBLP with appropriate restriction sites. This construct was digested with *Kpn*I/*Bam*HI and electroporated into wild type or *gefE*⁻ mutant cells to generate GFP knock-in recombinants.

### Ras activation assay

Vegetative or 12 hr filter developed cells were washed three times in Bonner's salts (10 mM NaCl, 10 mM KCl, 2 mM $CaCl_2$). Cells were lysed with 2 X lysis buffer (20 mM sodium phosphate, PH 7.2, 2% Triton X-100, 20% glycerol, 300 mM NaCl, 20 mM $MgCl_2$, 2 mM EDTA, 2 mM $Na_3VO_4$, 10 mM NaF, containing two tablets of protease inhibitor [Roche Complete] per 50 ml buffer). 400 µg of protein was then incubated with 100 µg of GST-Byr2-RBD on glutathione-sepharose beads at 4°C for 1 hr (*Bolourani et al., 2010*). The glutathione-sepharose beads were harvested by centrifugation and washed three times in 1 X lysis buffer. 50 µl of 1 X SDS gel loading buffer was then added to the pelleted beads and the suspension boiled for 5 min. Samples were subjected to SDS-PAGE and Western blots probed with anti-RasD specific antibody.

## Acknowledgements

This work was supported by a Lister Institute of Preventive Medicine Research Prize and Wellcome Trust Investigator Award (WT095643) to Chris Thompson.

## Additional information

### Funding

| Funder | Grant reference number | Author |
| --- | --- | --- |
| Wellcome Trust | 095643/Z/11/Z | Koki Nagayama, Lauren Harkin, Marzieh Kamjoo, Christopher RL Thompson |
| Lister Institute of Preventive Medicine | | Alex Chattwood, Koki Nagayama, Christopher RL Thompson |
| Natural Environment Research Council | NE/H020322/1 | Marzieh Kamjoo, Christopher RL Thompson |

| Funder | Grant reference number | Author |
|---|---|---|
| Royal Society Wolfson Merit | WM120109 | Christopher RL Thompson |
| Medical Research Council | G0900069 | Christopher RL Thompson |

The funders had no role in study design, data collection and interpretation, or the decision to submit the work for publication.

### Author contributions

AC, KN, PB, Conception and design, Acquisition of data, Analysis and interpretation of data, Drafting or revising the article; LH, MK, Acquisition of data, Analysis and interpretation of data; GW, CRLT, Conception and design, Analysis and interpretation of data, Drafting or revising the article

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
