## [Decision Letter]

Thank you for sending your work entitled “Developmental lineage priming by heterogeneous Ras activation” for consideration at *eLife*. Your article has been favorably peer reviewed by a Senior editor and two other reviewers.

The Senior editor and the other reviewers discussed their comments before we reached this decision, and the Senior editor has assembled the following comments to help you prepare a revised submission.

The reviewers all agreed that this work has provided novel insights into the mechanisms whereby lineage segregation can arise in a seemingly homogeneous population of cells. The aggregating slime mold system provides an interesting model to explore this mechanism, which has relevance in many other developmental systems. Previous work established that differentiation towards stalk or spore cells could be biased by the past history of the cells – cell cycle position or growth with or without glucose – and that this bias was manifest as altered sensitivity to the morphogen DIF. However, there were no clues as to the molecular basis of the bias. Your use of a novel genetic screen to identify GefE (a RasGEF) as a key downstream component in the pathway has opened up new insights into to the role of Ras activation levels in establishing cell heterogeneity and lineage specification. The reviewers do have some major concerns that would need to be addressed, most likely by additional experimentation, before we could reconsider the article for publication in *eLife*.

1) The selection for biased mutants is much more effective than expected. In Figure 1 around 20% of the input G-cells become spores (2% of 10%), and thus a mutant where all input cells become spores will only be enriched 5-fold per round of selection. If GefE^-^ was present as 1/3000 of the input cells (according to the size of the REMI library) then after 2 rounds of selection it could maximally only comprise 0.8% of the population. Yet according to one of the figures it is nearly 40%. This could be explained in 2 ways: i) the REMI library was very heavily biased to GefE^-^ cells; or ii) the selection does not work as expected. Perhaps spore germination in axenic medium is very asynchronous and GefE^-^ cells are selected at the start of each cycle because they hatch quickly?

There should be an attempt to resolve this conundrum. For instance a reconstruction experiment to show how efficiently GefE^-^ cells are selected; an idea of how complex the REMI library actually was (if possible) or whether repeat selections give different genes. Was GefE the only mutation isolated in the screen? Were multiple mutations in gefE isolated?

2) The heterogeneity of the RasD reporter is very interesting, but the chimera data appears somewhat confusing. Is the high RasD population making spore cells? Figure 7 seems to indicate that the high RasD population is good at making everything, but the low RasD population is excluded from the pre-stalk. Does Dif affect the probability that a cell is RasD positive or the extent to which it expresses RasD? How does this fit in with the binary ecm switch?

3) It was not clear why you think that variations in transcription of RasD could relate to cell fate, given that it is the levels of activated Ras that seem to relate to cell fate bias. Does this imply that RasD must be limiting in wild type cells and that it is converted to the ATP bound form upon translation? Perhaps this could be tested by Western blot? Or perhaps they could try over expressing RasD in wild type and GefE mutants, and assess the extent of rescue and levels of RasD-ATP? What happens to RasD protein levels? The authors seems to have a good antibody to RasD (not a given in this field) and so they could ask questions about protein levels: for instance whether cells sorted for high RasD expression actually have high RasD protein, or whether G- cells have more RasD than G+ cells.

[Editors’ note: the revised manuscript was re-reviewed, with the following revision requests.]

Thank you for resubmitting your work entitled “Developmental lineage priming by heterogeneous Ras activation” for further consideration at *eLife*. Your revised article has been favorably evaluated by a Senior editor and a member of the Board of Reviewing Editors. The manuscript has been improved and the reviewers favour publication. However, one reviewer has a remaining issue that should be addressed before acceptance, as outlined below:

With respect to the differences in the behavior of the RasD positive population the authors argue that the stability of the RFP explains the more promiscuous behavior of the RasD^+^ cells. This has also been observed in ESCs, as populations marked by fluorescent proteins have less defined functional activity. However, I am still a little confused about the dynamics of RasD response. The authors claim that RasD levels are DIF independent and this would be very interesting, but I am still confused as to what they think the mechanism is? They argue that RasD is only a part of early response, rather than the switch itself. How would this work? How does a RasD^-^ cell respond to Dif? Perhaps they could clarify these issues further.

---

## [Author Response]

*1) The selection for biased mutants is much more effective than expected. In*
Figure 1
*around 20% of the input G-cells become spores (2% of 10%), and thus a mutant where all input cells become spores will only be enriched 5-fold per round of selection. If GefE*^*-*^
*was present as 1/3000 of the input cells (according to the size of the REMI library) then after 2 rounds of selection it could maximally only comprise 0.8% of the population. Yet according to one of the figures it is nearly 40%. This could be explained in 2 ways: i) the REMI library was very heavily biased to GefE*^*-*^
*cells; or ii) the selection does not work as expected. Perhaps spore germination in axenic medium is very asynchronous and GefE*^*-*^
*cells are selected at the start of each cycle because they hatch quickly*?

*There should be an attempt to resolve this conundrum. For instance a reconstruction experiment to show how efficiently GefE*^*-*^
*cells are selected; an idea of how complex the REMI library actually was (if possible) or whether repeat selections give different genes. Was GefE the only mutation isolated in the screen? Were multiple mutations in gefE isolated*?

The referees are correct that the level of enrichment is higher than the 5x expected. Although it is impossible to accurately quantify the complexity of a REMI library (e.g., due to hot spots, etc.), we have performed several different experiments to resolve this issue as suggested. Firstly, we had already carried out a parallel selection on the same REMI library. Both selections resulted in an enrichment of mutants that can avoid the pstO and pstB cell bias imposed by glucose. We now include qualitative data showing this. Several mutants were isolated in both selections, however we were only able to confirm the gefE mutant through recapitulation experiments (i.e., the other mutants were caused by secondary mutations in addition to the identified REMI insertion). Most importantly, the *gefE* mutant was detected in both libraries, but at round 2 and round 6. We believe that these differences are likely due to the effects of random genetic drift caused by harvesting a finite number of spores after each round. Alternatively, it has recently been shown that when multiple clones are hatched and grown on bacteria, small differences in growth rate at the feeding edge can be amplified, leading to overrepresentation of some clones (9). We have modified the text in the Results to include this.

Furthermore, as suggested, we have performed reconstruction experiments to verify that enrichment of *gefE*^-^ mutant cells is due to overrepresention in the spore population, rather than the mutation simply conferring advantages during other stages of the selection process (e.g., spore hatching, growth, etc). These reconstruction experiments revealed no difference in the rate of growth of *gefE*- mutant cells in either G+ or G- medium. However, *gefE*^-^ mutant spores did hatch at a faster rate than wild type spores, perhaps also helping to explain the unexpectedly strong enrichment for this mutant within the pools. These data are now included and are described in the Results section. We have provided a revised estimate of the REMI library composition based on these observations.

*2) The heterogeneity of the RasD reporter is very interesting, but the chimera data appears somewhat confusing. Is the high RasD population making spore cells?*
Figure 7
*seems to indicate that the high RasD population is good at making everything, but the low RasD population is excluded from the pre-stalk*.

The reviewer is correct to point out that as ‘low RasD’ cells (as measured by rasD^promoter^ driven RFP expression) make only prespore cells then cells expressing high *rasD* transcript levels might be expected to make only prestalk cells. The observation that cells expressing high *rasD* transcript levels can form both prespore and prestalk cells, as evidenced by their random distribution at both slug and culminant stages is thus surprising. We believe that these results are still consistent with the idea that there is an increased likelihood that high RasD cells will become prestalk cells compared to low RasD cells. One likely explanation for their failure to solely become prestalk cells is that the stability of the RFP protein may be different to endogenous RasD. For example, if RasD protein has a shorter half life than RFP, then not all cells that express RFP would actually express RasD. Consequently, not all RFP expressing cells would exhibit a prestalk bias and may be free to differentiate as prespore cells, thus hindering experimental observation of their behavior. One observation supporting this idea is that the low RasD expressing population exhibits slightly faster dynamics than the high RasD expressing population, thus supporting the idea that chimeric behavior may be affected by differences in RasD transcription/translation and RFP stability. We have now included expanded description and discussion of these ideas to clarify these issues within the Results and Discussion.

*Does Dif affect the probability that a cell is RasD positive or the extent to which it expresses RasD? How does this fit in with the binary ecm switch*?

Consistent with previous studies (33), we find that *rasD* transcription is unaffected by DIF treatment, thus suggesting that positive feedback through *rasD* transcriptional activation does not contribute to the binary switch ((Jermyn, 1987) and data not shown). Consequently, it seems likely that Ras only acts during the immediate-early phase to set up DIF responses. We have now included a discussion of these observations in the Discussion.

*3) It was not clear why you think that variations in transcription of RasD could relate to cell fate, given that it is the levels of activated Ras that seem to relate to cell fate bias. Does this imply that RasD must be limiting in wild type cells and that it is converted to the ATP bound form upon translation? Perhaps this could be tested by Western blot? Or perhaps they could try over expressing RasD in wild type and GefE mutants, and assess the extent of rescue and levels of RasD-ATP? What happens to RasD protein levels? The authors seems to have a good antibody to RasD (not a given in this field) and so they could ask questions about protein levels: for instance whether cells sorted for high RasD expression actually have high RasD protein, or whether G- cells have more RasD than G+ cells*.

The reviewer is correct that our model for the action of GefE and RasD assumes that the levels of RasD transcription are limiting. Based on their suggestions, we have now performed several additional experiments to strengthen the support for this important idea:

A) We have measured the levels of total and activated RasD in G+ and G- grown cells (it is impossible to prepare sufficient material from FACS samples). Consistent with the idea that RasD transcription is induced in G- medium, we find that the levels of total RasD protein are also significantly increased. Most importantly, we find that the levels of activated RasD are also similarly increased. These new findings are now described in the results section and presented in Figure 6.

B) We have compared the effects of overexpressing wild type RasD and constitutively active RasD(G12T). In wild type cells, their effects are almost indistinguishable, resulting in a bias towards prestalk fates, suggesting that RasD transcription is limiting. Most importantly, however, this effect is only seen when constitutively active RasD(G12T) is expressed in *gefE*^*-*^ mutant cells, suggesting that GefE is required to activate the overexpressed RasD. These new findings are now described within the results section and presented in Figure 6.

[Editors’ note: the revised manuscript was re-reviewed, with the following revision requests.]

*With respect to the differences in the behavior of the RasD positive population the authors argue that the stability of the RFP explains the more promiscuous behavior of the RasD*^*+*^
*cells. This has also been observed in ESCs, as populations marked by fluorescent proteins have less defined functional activity*.

We have interpreted the observation of unexpected lineage promiscuity of RasD RFP positive cells as being due to stability of RFP protein, We have interpreted the observation of unexpected lineage promiscuity of RasD RFP positive cells as being due to stability of RFP protein. However, we are grateful to the reviewer for pointing out that such differences in behavior have also been observed in ESC cultures and the early mouse embryo, where this explanation has not been proposed. We have now noted this interesting parallel within the Discussion. Furthermore, we note the importance of the generation of faithful real time reporter genes to truly understand how differences in expression impact on lineage choice.

Surprisingly, however, we found that cells expressing high *rasD: RFP* transcript levels only exhibit a relatively weak prestalk preference. One interpretation of this finding is that if RasD protein has a shorter half life than RFP, then not all cells that express RFP would actually express RasD. Consequently, not all RFP expressing cells would exhibit a prestalk bias and may be free to differentiate as prespore cells, thus hindering experimental observation of their behavior. It is interesting to note, however, that similar observations of cell populations behaving more promiscuously in their lineage restriction than might be expected, have also been seen when fluorescent protein reporters are used in studies of the early mouse embryo and in embryonic stem cell cultures (Canham et al., 2010; Grabarek et al., 2012; Morgani et al., 2013). Most importantly, all these observations highlight the need for faithful real time reporters of gene expression, which will permit future studies of the regulation of RasD promoter activity, the molecular basis of transcriptional heterogeneity and lineage priming in general.

*However, However, I am still a little confused about the dynamics of RasD response. The authors claim that RasD levels are DIF independent and this would be very interesting, but I am still confused as to what they think the mechanism is? They argue that RasD is only a part of early response, rather than the switch itself. How would this work? How does a RasD- cell respond to Dif? Perhaps they could clarify these issues further*.

To clarify these issues, we have:

A) Included a much clearer description of the evidence that RasD levels fluctuate during growth, when DIF levels are negligible. We also further state that DIF has no effect of RasD transcription during growth or development. Based on these observations, we now include a full discussion of the putative regulators of RasD expression during growth. For example there is extensive literature describing how growth heterogeneities (e.g., cell cycle position, intracellular calcium concentration) affect lineage choice. However, the molecular mechanism underlying these heterogeneities, or how they affect lineage choice is currently unknown. We therefore agree that a discussion of these earlier studies is important in the context of our findings. Most importantly, this will likely provide the intellectual framework for future studies into the molecular basis linking these heterogeneities to transcriptional heterogeneity (e.g., RasD) and thus lineage choice.

B) Included a clear discussion of how we believe RasD might feed into the DIF signaling pathway to affect DIF responses. We now fully describe the evidence supporting the idea that RasD is not a central component of the DIF signaling pathway and thus only affects the sensitivity of cells to the DIF signal (i.e., tunes DIF responsiveness). Consequently RasD deficient cells are ultimately able to respond to DIF when given prolonged exposure to the DIF signal. We thus provide a full discussion of the putative mechanism(s) by which RasD could affect the activity of the DIF signaling pathway. This is important, as it has enabled us to highlight what is known about the DIF signaling pathway, as well as allowing us to emphasize the gaps in our understanding. It is hoped that this will stimulate further research in this area, as it is only through understanding all components of a signaling pathway and their interactions will we be able to understand how subtle tuning of sensitivity is achieved at a molecular level.

Immediate-early DIF responses have been shown to be digital, with the number of responding cells being strongly affected by changes in DIF concentration, whereas the magnitude of the response in individual cells is largely unaffected (61). This observation implies that 1) a response is only triggered once an internal threshold is exceeded and 2) the position of the threshold varies across a population. Work described in this report provides further support for this idea and gives mechanistic insight into how the DIF response threshold is set and, thus, how cell fate choice is regulated in vivo.

Specifically, we find that GefE dependent heterogeneous RasD activation during the growth phase is required to set the normal DIF response threshold. However, several major questions emerge. For example, DIF is a developmental inducer that is present only after cells have been starved; yet the DIF response threshold is set (by RasD levels) before DIF exposure, in the growth phase. Indeed, heterogeneity in *rasD* expression is independent of DIF, as RasD expression is unaffected by DIF treatment ((33) and data not shown). One clue to how this heterogeneity is generated comes from our observation that RasD protein levels are higher in cells grown in medium lacking glucose. This suggests that heterogeneity in metabolic state between cells may play a role in *rasD* regulation. However, other growth phase heterogeneities, including differences in cell cycle position (25; 4; 63), intracellular calcium concentration (6; 5) and intracellular pH (27; 43) have all also previously been shown to affect cell fate choice. In addition, the molecular basis underlying these heterogeneities, or whether they affect RasD expression, is currently unknown. It thus remains to be determined whether they are a cause or consequence of the observed differences in RasD levels.